# Calibration of Mobile Robots Using ATOM

**DOI:** 10.3390/s25082501

**Published:** 2025-04-16

**Authors:** Bruno Silva, Diogo Vieira, Manuel Gomes, Miguel Riem Oliveira, Eurico Pedrosa

**Affiliations:** 1Institute of Electronics and Informatics Engineering of Aveiro (IEETA), University of Aveiro, 3810-193 Aveiro, Portugal; diogoscsv@ua.pt (D.V.); manuelgomes@ua.pt (M.G.); mriem@ua.pt (M.R.O.); efp@ua.pt (E.P.); 2Department of Mechanical Engineering, University of Aveiro, 3810-193 Aveiro, Portugal

**Keywords:** extrinsic calibration, intrinsic calibration, registration, multi-modal, multi-sensor, ROS

## Abstract

The calibration of mobile manipulators requires accurate estimation of both the transformations provided by the localization system and the transformations between sensors and the motion coordinate system. Current works offer limited flexibility when dealing with mobile robotic systems with many different sensor modalities. In this work, we propose a calibration approach that simultaneously estimates these transformations, enabling precise calibration even when the localization system is imprecise. This approach is integrated into Atomic Transformations Optimization Method (ATOM), a versatile calibration framework designed for multi-sensor, multi-modal robotic systems. By formulating calibration as an extended optimization problem, ATOM estimates both sensor poses and calibration pattern positions. The proposed methodology is validated through simulations and real-world case studies, demonstrating its effectiveness in improving calibration accuracy for mobile manipulators equipped with diverse sensor modalities.

## 1. Introduction

The fusion of data from multiple sensor sources is often based on the premise that accurate geometrical transformations between those sensors are known. The methodology employed to determine these transformations is called extrinsic calibration, and it is therefore considered to be a core component of multi-modal sensor fusion [1]. Numerous applications of robotic systems require accurately calibrated sensor ensembles, such as Simultaneous Localization and Mapping (SLAM) [2,3], structure from motion [4], 3D reconstruction [5], people tracking [6], motion capture REF, autonomous driving [7], among many others. This results in large, heterogeneous sensor sets, from simple RGB-D camera systems [8], to stereo cameras [9], to more advanced systems such as intelligent vehicles [10,11,12] or multi-sensor industrial cells [13]. This vast set of heterogeneous configurations sprawls into different calibration problems, which have been addressed separately in the literature.

Oliveira et al. [14] categorized calibration problems into three classes. The first, called the **sensor to sensor calibration problem**, is the classical case where the goal is to estimate the transformation of one sensor w.r.t. another. A second case, referred to as the **sensor in motion calibration problem**, aims to estimate the geometric transformations between the poses of one sensor in motion. One clear example involves structure from motion applications using bundle adjustment methodologies [15]. Finally, the third case, entitled the **sensor to coordinate frame calibration problem**, refers to calibrations where the objective is to estimate a geometric transformation between a sensor and a coordinate frame of interest. In this case, the sensor is required to move in order to collect data from different viewpoints. One clear example is hand–eye calibrations [16,17,18], where the goal is to estimate the pose of a camera w.r.t. the end effector coordinate system. In addition to this, there are multiple sub-variants of these problems, generated by the number of sensors as well as their modalities. As a result, there are currently many available calibration tools, but most of them operate under the paradigm of a specific calibration problem, are limited to specific sensor modalities, and often calibrate only a pair of sensors.

Calibrating an autonomous mobile manipulator poses a multifaceted challenge, combining the intricacies of mobile robots with the additional complexities introduced by robotic manipulators. The calibration process encompasses several pivotal aspects. First and foremost, it necessitates the calibration of the onboard sensors on the mobile manipulator in relation to each other, giving rise to the fundamental challenge of a **sensor to sensor** calibration problem. This calibration ensures that the sensors aboard the manipulator provide consistent and precise measurements, establishing a foundation for reliable perception and control. Moreover, owing to the mobility of both the mobile platform and the manipulator itself, it becomes imperative to estimate the transformations between the sensor poses over time. This introduces a dynamic facet into the calibration process, the **sensor in motion** component.

This element is inherently entwined with the motion coordinate system of the vehicle, requiring the estimation of the geometric transformation between the sensors and the motion coordinate system, constituting the **sensor to coordinate frame** component. Additionally, the manipulator’s capability to manipulate objects hinges on precise hand–eye coordination. This entails establishing a clear understanding of the geometric transformation between the sensors (typically cameras or vision systems) and the manipulator’s coordinate frame. This element also encompasses a sensor to coordinate frame calibration problem, which is indispensable for aligning sensory perception with the manipulator’s motion control, ensuring precise interaction with the environment. Beyond sensor calibration, manipulator joint calibration is also crucial for hand–eye systems. Joint misalignments can propagate errors into extrinsic sensor calibrations in these systems [19].

Furthermore, contemporary robotic systems are evolving to incorporate a multitude of sensors with various modalities. This trend aims to leverage multiple sensors to offer diverse perspectives of the environment. The integration of multi-sensor and multi-modal data collection is pursued to enhance system robustness, collecting data of varying natures to ensure resilience in challenging scenarios. This integration of further modalities amplifies the intricacies of the calibration process.

ATOM (https://github.com/lardemua/atom, accessed on 1 February 2025) [14] is a calibration framework able to tackle a large set of calibration problems. It has been successfully applied to calibrate very different systems, ranging from hand–eye systems [16] to industrial collaborative cells [13]. ATOM was also used to calibrate autonomous vehicles [20] as well as automated guided vehicles for agriculture [21]. However, the calibrations of these vehicles were incomplete, since they produced only either the sensor to sensor transformation estimates, or, in the case of the hand–eye systems, the sensor to frame transformation estimates. In other words, these calibrations were simplified to consider only one type of calibration problem.

This paper describes the work carried out to extend the ATOM calibration framework, so that it is able to estimate the poses of the sensors onboard mobile manipulators, as well as those of other relevant coordinate systems such as the vehicle motion coordinate frame and the manipulator coordinate frame. The contributions of this paper are to present a calibration method that:estimates the transformation between the sensors and the motion coordinate system;estimates the transformations given by inaccurate localization systems, avoiding the negative impact that these transformations could have on the calibration accuracy;can calibrate with accuracy a complex real mobile manipulator, with many sensor modalities.

## 2. Related Work

Various authors have endeavored to address the challenge of sensor calibration for mobile manipulators. Ref. [22] devised an optimization-based calibration approach utilizing arbitrary trihedrons to calibrate an RGB camera attached to the end-effector of the manipulator and a Light Detection And Ranging Sensor (LiDAR) sensor on the mobile platform. Trihedrons are commonly found in most indoor environments, typically at the corners of rooms. Notably, during the calibration process, the mobile platform remains stationary while the manipulator is in motion, simplifying the problem to a hand–eye calibration scenario.

Ref. [23] introduced a calibration technique to calibrate two 2D LiDAR sensors using a room corner as a reference point. Both LiDAR sensors are mounted on the mobile platform, with no sensors attached to the manipulator, which reduces the problem to a more manageable state.

Ref. [24] proposed an approach for calibrating an RGB camera attached to the mobile platform concerning the end-effector, incorporating contact-based interaction with a force-torque sensor on the manipulator. This approach does not address a crucial transformation for mobile manipulators, namely the transformation between the mobile platform and the manipulator. The system also lacks a sensor on the end-effector, reducing the complexity of the problem.

Ref. [25] introduced an intriguing method, which involves calibrating two RGB cameras on the mobile platform of a mobile manipulator concerning the base of the manipulator. This is made possible by the presence of a projector mounted on the end-effector of the manipulator, which projects a pseudorandom coded light that functions as a calibration pattern. Notably, similarly to the system used in the approach by [24], this system lacks a sensor on the end-effector, reducing the complexity of the problem.

Each of these approaches is highly specialized for its specific context and may not readily generalize to new systems with different sensor configurations, multiple sensors, and various modalities.

Extrinsic calibration is as a procedure that estimates the geometric transformation from one sensor to another, to a coordinate frame of interest, or even from a sensor to itself when in motion. This procedure is always a subproblem inherent to the calibration of mobile manipulators. Whenever the transformation from a sensor to another is being estimated, we refer to this as pairwise sensor calibration and argue that these are not adequate solutions to calibrate most robotic mobile systems. Pairwise sensor calibration can be carried out with a single sensor modality, such as RGB to RGB [26,27,28,29], LiDAR to LiDAR [30,31], etc., or between various modalities, such as RGB to LiDAR [32,33,34] or RGB to Depth camera [35,36,37].

Generalizing pairwise calibration to systems with more than two sensors is not straightforward, as it necessitates additional pairwise procedures for all possible sensor combinations in the system [14]. To address this, a common solution is the use of sequential pairwise approaches, where several pairwise calibrations are conducted and organized into a graph-like sequential procedure. In this approach, each sensor is calibrated in relation to the next one in a given order, forming a sequential path of transformations.

This results in a topological representation of the system, where nodes represent sensor coordinate systems, and edges represent the estimated transformations between them. By ensuring that the topological representation is connected, it becomes possible to compute transformations between any pair of sensors by retrieving the topological path and combining the corresponding pair-wise calibration estimates. Nonetheless, the pairwise paradigm is suboptimal as it results in error propagation, underutilizes the available data within the system, and becomes unwieldy when applied to systems with a higher quantity of sensors.

Additionally, pairwise calibration approaches assume rigidly attached sensors, making them unsuitable for scenarios where a sensor may move in relation to another during calibration, such as in sensor-in-motion calibration or sensor to frame calibration problems [14]. From this, it can be concluded that pairwise calibration is unfeasible for more complex robotic systems, such as mobile manipulators.

As mentioned before, one subproblem inherent to the calibration of mobile manipulators is the hand–eye calibration problem, an extrinsic calibration problem. Hand–eye calibration is a classical problem [38] that consists of estimating the transformation between the end-effector of a robotic manipulator and a camera mounted on top of that end-effector. The most well-known formulations of this problem include those using relative geometric transformations and absolute geometric transformations.

The first formulation uses different transformations for the same frame of reference in different instances of time. It can be formulated as:(1)AX=XB,
where A and B are the known transformations between various timesteps of the end-effector and the camera, respectively, and X is the unknown transformation between the end-effector and the camera [39].

The second formulation is:(2)AX=ZB,
where A and B are the known transformations between the frames of the end-effector and the robot base and the frames of the camera and the pattern, respectively, while X and Z are the unknown transformation between the robot base and the pattern and the unknown transformation between the end-effector and the camera, respectively [40].

While these formulations are interesting for solving the hand–eye problem, they are limited in their ability to generalize to the case of mobile manipulators. Ref. [41] introduced a new formulation optimized for mobile manipulators:(3)AX=YB,
where A and B are the known transformations between the frames of the mobile platform and the world and the frames of the camera and the pattern, respectively, while X and Y are the unknown transformation between the world and the pattern and the unknown transformation between the mobile platform and the camera, respectively. This formulation creates a problem as Y comprises two unknown transformations: one between the mobile platform and the base of the manipulator and one between the end-effector and the camera. Although the camera is calibrated with respect to the mobile platform, it is not with respect to the end-effector. This limits its applicability, particularly in scenarios requiring accurate grasping. Additionally, it remains confined for systems with only a single camera in the end-effector.

## 3. Materials and Methods

This section presents a general methodology applicable across various robotic systems. While not explicitly tailored to robotic manipulators, the approach incorporates enhancements that enable their calibration as a natural extension of the framework. Later, in Section 4, a case study of a real mobile manipulator is showcased, alongside two other simpler case studies. This section first formulates the problem in Section 3.1; secondly, in Section 3.2, our approach is detailed, and from Section 3.3 and onwards, the cost functions developed are introduced for the various supported sensor modalities.

### 3.1. Problem Formulation

The classic paradigm for designing calibration cost functions is to establish a cost function to measure the quality of the estimate of the transformation between two sensors $s_{a}$ and $s_{b}$, T^sbsa. The calibration pattern is detected in the data of one sensor $s_{b}$, and then projected to the other sensor $s_{a}$. The cost is then obtained by assessing the difference between the projected pattern and the pattern as detected in the data of the second sensor, which can be generically written as:(4)argminT^sbsa∑c∈C∑d∈De{m(sa),m(sb)}dc,sa,dc,sb,T^sbsa,{λsa},{λsb},
where $d_{c,s_{␣}}$ is a detection of a key-point in the pattern for an arbitrary sensor, namely $s_{a}$ and $s_{b}$, e.g., a chessboard corner for a given collection *c* and sensor *s*, D is the set of detections, e(·) is a generic error function that compares the detections in one sensor $d_{c,s_{a}}$ to corresponding projections obtained from the detections of other sensor $d_{c,s_{a}}$. We use the term *collections* to denote the specific moments in time in which data are collected from all onboard sensors. In addition to this, each collection also stores all the geometric transformations available between the coordinate frames of the system. In the calibration pipeline, we refer to a dataset as a set of collections. To carry out these projections, we need the geometric transformation between the sensors T^sbsa, and possibly some additional parameters from each sensor, as in the case of the intrinsic parameters of RGB sensors, which are represented by $\left\{\lambda_{s_{a}}\right\}$ and $\left\{\lambda_{s_{b}}\right\}$. Note that e(·) differs according to the combination of modalities of the sensor pair, denoted by $\left\{m\left(s_{a}\right),m\left(s_{b}\right)\right\}$, where m(·) is a function that retrieves the modality of the sensor. Finally, *c* denotes a collection from the set of collections C, to account for the fact that accurate calibrations must take into account several snapshots of data.

We refer to this as a **sensor to sensor calibration paradigm.** The problem with this methodology is that it is not scalable, in particular in the case of complex robotic systems, which contain several sensors. In these situations, accounting for all the pairwise sensor combinations may become cumbersome [14], and transforms (Equation 4) into:(5)argmin{T^}∑{sa,sb}∈S∑c∈C∑d∈De{m(sa),m(sb)}dc,sa,dc,sb,T^sbsa,{λsa},{λsb},
where $\left\{\hat{T}\right\}$ denotes the set of transformations to be optimized, and $\left\{s_{a},s_{b}\right\}$ is the pair of sensors $s_{a}$ and $s_{b}$, extracted from the set of all sensor pairwise combinations S. Note that this pairwise formulation requires the design of a specific error function for each pairwise combination of sensor modalities. This implies that an error functions must be designed to tackle pairs of RGB cameras, $e_{\left\{rgb,rgb\right\}}\left(\cdot \right)$, RGB camera with LiDAR, $e_{\left\{rgb,lidar\right\}}\left(\cdot \right)$, a pair of LiDARs, $e_{\left\{lidar,lidar\right\}}\left(\cdot \right)$, etc.

The proposed approach offers an alternative way of designing these cost functions, which considerably simplifies both their design and computation. The first step is to add to the optimization an estimate of the pose of the calibration pattern. With this, it is possible to design the cost function by projecting the estimated pose of the calibration pattern to a single sensor, and measuring the difference between said projection and the detection of the pattern in the data of that sensor. This can be formulated as:(6)argmin{T^}∑s∈S∑c∈C∑d∈Dem(s)dc,s,d{p},T^ps,{λs},
where S is the set of sensors, $d_{\left\{p\right\}}$ denotes the location of the key-points that correspond to the detections $d_{c,s}$, defined in the coordinate frame of the pattern, and computed after the known parameterization of the calibration pattern which is described in the set of parameters {p}. These parameters may include the number of horizontal and vertical squares, in the case of chessboard patterns [42], or the dictionary of markers for charuco boards [43]. In order to project $d_{\left\{p\right\}}$ from the pattern *p* to the sensor *s* coordinate frames, we now need transformation T^ps.

We argue that this **sensor to pattern calibration paradigm** presents several benefits in comparison to the classic approaches. It has the advantage of tackling each sensor data independently by avoiding pairwise arrangements, and considerably simplifying the calibration computations. This can be seen by comparing the expression for the classic approaches (Equation 5), with the one for the proposed approach (Equation 6). The procedure is carried out for each sensor *s*, in the set of sensors S, rather than accounting for all pairwise combinations $\left\{s_{a},s_{b}\right\}$ in the set S, as in (Equation 5). Additionally, the error function $e_{m(s)}\left(\cdot \right)$ does not depend on the combination of modalities, but rather on the modality of one sensor. This reduces the number of error function variants to be designed. Each sensor is tackled separately, therefore simplifying the design of each error function. Nonetheless, the geometric relationships between all the sensors in the system are still taken into account, since the estimated location of the pattern creates an indirect connection between sensors. These benefits come at the cost of augmenting the optimization problem, through the additional estimation of the pose of the pattern, but this has no significant impact in the speed or accuracy of the optimization.

### 3.2. Optimization of Atomic Transformations

Equation (Equation 6) describes the proposed approach as a sensor to pattern formulation, because the pattern is projected into the coordinate frame of each sensor. This is carried out by estimating T^ps, the transformation from the sensor *s* to the pattern *p*. However, it is important to highlight that T^ps represents, in most cases, an aggregate transformation. In other words, it means that it is possible to disassemble T^ps into smaller components. Consider the example of a sensor assembled on the end effector of a robotic arm which is mounted on a mobile platform; in this case, the transformation from the sensor to the pattern will include several sub-transformations, such as the transformations from the sensor to the end effector, from the end effector to the base of the manipulator, from the manipulator to the base of the mobile platform, and finally, from the base of the mobile platform to the calibration pattern. Moreover, these sub-transformations can be subdivided further. For example, the transformation from the end effector to the base of the robotic arm can be defined as the product of the transformations between the base, arm, forearm, wrist, etc. However, these sub-transformations of T^ps may be very different. While some may be known with high accuracy, others could have poor estimations. Some may be dynamic, while others are static throughout the procedure. The estimation of the aggregate transformation T^ps does not take into account any of this.

We propose to carry out an optimization procedure that excludes all aggregate transformations from the optimization procedure, keeping only the indivisible transformations. We refer to these as **atomic transformations.** To put together the atomic transformations, a topology with the connections between coordinate frames must be defined, where nodes represent coordinate frames, and edges denote the transformations between these frames. This is commonly known as a transformation tree [44]. Given a known topology between coordinate frames, the aggregate geometric transformation between any two coordinate frames $f_{a}$ and $f_{b}$ in the system can be computed as:(7)Tcfbfa=∏fn∈fa⇾fbTcfn+1fn,
where $f_{a}⇾f_{b}$ denotes the topological path from $f_{a}$ to $f_{b}$, i.e., the sequence of coordinate frames visited when traveling from $f_{a}$ to $f_{b}$, and Tcfn+1fn denotes the atomic transformation from frame $f_{n}$ to $f_{n+1}$. Note that $f_{n}$ and $f_{n+1}$ must be topological neighbors, to ensure that the transformation is atomic. The sub-index *c* is present to account for the fact that these transformations may be dynamic, i.e., could change in value from collection to collection.

The proposed method is called Atomic Transformations Optimization Method (ATOM), since the idea is to use a single optimization procedure to jointly estimate all atomic transformations marked to be calibrated. It can be formulated as:(8)argmin{T^}∑s∈S∑c∈C∑d∈Dem(s)dc,s,d{p},∏fn∈s⇾pTcfn+1fn,{λs},
where $\left\{\hat{T}\right\}$ is the list of estimated atomic transformations. When estimating a dynamic transformation, the transformation for each collection is estimated instead. This happens, for example, when calibrating a static robot with a moving pattern. The reasoning behind this is that if the topological path from any sensor to the pattern, s⇾p, contains a dynamic transformation to be estimated, like the transformation from the world to a moving pattern, and only the transformation of the first collection is estimated, the cost function for subsequent collections will not be computed correctly and lead to bad calibration results. This formulation is general since it can be applied to any robotic system.

Moreover, as will be detailed in Section 4.2, it is also very flexible, since the calibration can be configured by marking different atomic transformations to be estimated. Previously, in our previous work, Ref. [14], it was assumed that multiple marked transformations on the same topological path from a sensor to the pattern, s⇾p, would lead to ambiguities and unwanted convergence on local minima. However, this was found to not be the case when these transformations are present on the referred topological paths of other sensors as well. By sharing these additional transformations amongst many sensors, the potential for ambiguity is offset by having a wide variety of data from many points of view. This led to the addition of what we named *additional transformations*, an additional set of arbitrary atomic transformations that can be estimated, alongside the previous atomic transformations to be estimated, $\left\{\hat{T}\right\}$. These transformations can be anything, from the transformation given by a localization system, to the transformation from a base of a platform to a robotic manipulator. This finding was the key to unlocking the calibration of mobile manipulators.

### 3.3. RGB Modality

Section 3.1 and Section 3.2 have described ATOM as a sensor to pattern calibration methodology based on the optimization of atomic transformations. The proposed approach is expressed in (Equation 8). This is a generic expression, in which the error function $e_{m(s)}$ will vary according to the sensor modality. In the case of the RGB modality, the error function $e_{rgb}$ is defined as:(9)ergb=xd,c,s−P∏fn∈s⇾pTcfn+1fn·xd,c,pxyz,{ks},{us}F2,
where $x_{c,s,p}$ denotes the three-dimensional homogeneous coordinates of the pattern corner that correspond to detection *d*, which has pixel coordinates $x_{c,s,d}$ in the image of sensor *s* at collection *c*, ${\left[\cdot \right]}_{xyz}$ is an operator that removes the homogeneous coordinate, $P\left(\cdot \right):R^{3}\to R^{2}$ is the pinhole camera model projection function that makes use of the set of intrinsic $\left\{k_{s}\right\}$ and distortion $\left\{u_{s}\right\}$ parameters included in $\left\{\lambda_{s}\right\}$, and finally, ${∥\cdot ∥}_{F}^{2}$ is the Frobenius norm.

### 3.4. Range Modalities

The cost function for the range sensor modalities includes both LiDARs and depth camera sensors. The idea is to compare the pose of ground truth key-points, defined in the pattern’s coordinate frame, with sensor detections, which are transformed into the pattern’s coordinate frame. The function evaluates two components: one that captures error in the direction orthogonal to the plane of the pattern, and another that works in the longitudinal direction. The reason for splitting the error function into these two components is that it opens the possibility of using different sets of detection key-points for each. As such, the orthogonal error is estimated from all the measurements from the LiDAR sensor that were labeled as belonging to the pattern, while the longitudinal error can only be computed from the subset of key-points located at the boundaries of the pattern. The orthogonal error $e_{range,\;o}\left(\cdot \right)$ is formulated as:(10)erange,o=∏fn∈p⇾sTcfn+1fn·xd,c,sz
where $x_{d,c,s}\in R^{3}$ are the coordinates of the LiDAR points labeled as pattern detection, defined in the sensor’s coordinate frame, and ${\left[\cdot \right]}_{z}$ is an operator that extracts the *z* coordinate, the only useful coordinate for assessing the orthogonal error. Note that, unlike in (Equation 9), here it is more adequate to transform the sensor measurements to the pattern’s coordinate frame. This is why, in this case, we use the inverse topological path p⇾s.

Let the set of sensor measurements which are labeled as the pattern and that are located near the boundary of the pattern be denoted as B, and let Q be the set of points located at the boundary of the pattern, extracted from the known parameters of the pattern. The longitudinal error $e_{range,\;o}$ operates by assessing, for each boundary detection d∈B, which is the closest ground truth point q∈Q. The procedure is formulated as:(11)erange,l=minq∈Qxq,c−∏fn∈p⇾sTcfn+1fn·xb,c,sxyF2,
where $x_{q,c}$ is the 3D coordinate of the ground truth boundary point, defined in the pattern’s coordinate frame, $x_{b,c,s}$ denotes the 3D coordinate of the sensor measurement, defined in the sensor’s coordinate frame, and ${\left[\cdot \right]}_{xy}$ is an operator that extracts the *x* and *y* coordinates, i.e., those related to the longitudinal displacement.

## 4. Results and Discussion

### 4.1. Calibration Metrics

This section provides a detailed account of the metrics employed for assessing the performance of calibration methods. RGB to RGB evaluation metrics and metrics to compare results to the ground truth are thoroughly explained. Evaluation metrics pertaining to other sensor modalities such as depth or LiDAR can be consulted in our previous work, Ref. [14]. Most evaluations in this study adopt a pairwise approach, which is consistent with the predominant nature of calibration techniques. This pairwise evaluation framework facilitates meaningful comparisons between different calibration methodologies.

#### 4.1.1. RGB to RGB Evaluation

The RGB to RGB evaluation involves three distinct error parameters: mean rotation $\epsilon_{R}$, mean translation $\epsilon_{t}$, and root-mean-square $\epsilon_{\mathrm{rms}}$. These metrics are utilized to assess the disparities between the pattern poses detected by each sensor. This assessment is facilitated by estimating the transformations between the RGB cameras and the pattern, denoted as T^cps, using a perspective-N-point methodology [45,46].

However, a direct comparison between these transformations from different sensors is not straightforward, as the transformations of the pattern must be expressed with respect to a common frame. To achieve this, we rely on the atomic transformations in the topological path between the sensors and the world, which have been either predefined or estimated through a calibration approach. This can be expressed as(12)T[s,c]pw=∏fn∈w→sTcfn+1fn·sT^cp.

With this transformation established, two transformations, T[sa,c]pw and T[sb,c]pw, one for each sensor, can be determined. It can be stated that, by definition, these transformations should be equal; hence,(13)T[sa,c]pw=T[sb,c]pw.

This equation can be further expanded as(14)R[c,sa]pwt[c,sa]pw01=R[c,sb]pwt[c,sb]pw01,
where R[c,s␣]pw and t[c,s␣]pw represent, respectively, the 3 × 3 rotation matrix and the 3 × 1 position vector extracted from a homogenous transformation matrix T[c,s␣]pw, for a given sensor. From this, we can define the mean rotation error as(15)ϵR=1|C|∑c∈C||angle((Rc,sapw)−1·Rc,sbpw)||,
where C represents the set of all collections, and angle(·) denotes the angle-axis representation of the rotation, obtained by extracting the *Rodrigues* angles from the 3×3 rotation matrix. The mean translation error is calculated as(16)ϵt=1|C|∑c∈C‖tc,sapw−tc,sbpw‖.

To calculate the root-mean-square error, the pixel coordinates of a detection *d* of the pattern on sensor $s_{b}$ ($u_{\left[c,s_{b},d\right]}$) is compared with the projection of the pixel coordinate of the same detection *d* of the pattern on sensor $s_{a}$ ($u_{\left[c,s_{a},d\right]}$) to sensor $s_{b}$ ($x_{\left[c,s_{a}\to s_{b},d\right]}$), across all detections within a set of detections D representing the corners of the calibration pattern. The projections can be expressed as(17)x[c,sa→sb,d]=Ksb·T[sb,c]pw·(T[sb,c]pw)−1·(Ksa)−1·u[c,sa,d],
where $K_{s}$ represents the intrinsic parameters of sensor *s*, including the focal length parameters, $f_{x}$, $f_{y}$, and the optical center point parameters, $c_{x}$, $c_{y}$ [47]. The root-mean-square error is then calculated as(18)$$\epsilon_{\mathrm{rms}}=\frac{1}{\left|C\right|}\sum_{c\in C}\sqrt{\frac{1}{\left|D\right|}\sum_{d\in D}\|u_{\left[c,s_{b},d\right]}-x_{\left[c,s_{a}\to s_{b},d\right]}{\|}_{F}^{2}}.$$

#### 4.1.2. Ground Truth Comparison

The quality of the calibration of the transformations between frames without sensorial data cannot be assessed directly. The transformation between the world and the base of a robot is one example. These metrics provide unambiguous insights, but they are not transposable to real-world systems and are thus often less desirable. In simulated environments, the calibration performance of these transformations can be evaluated by extracting the transformation from the simulator before adding artificial noise and comparing the final estimation with the ground truth. If two transformations are equal, then:(19)$$T_{a}\cdot {\left(T_{b}\right)}^{-1}=I_{4},$$
where $T_{a}$ and $T_{b}$ are the two supposedly equal transformations and $I_{4}$ is a identity matrix of fourth order. The remaining translation and rotation components are used to compute the translation error, $\epsilon_{t}$, and rotation error, $\epsilon_{R}$. The translation error is computed from the norm of the position vector encoded in the last column of the resulting matrix, while the rotation error is computed from the norm of rotation vector with Euler angles, extracted from the rotation sub-matrix.

### 4.2. Case Studies

This section entails the case studies conducted to prove how the proposed methodology solves the main challenges revolving around calibrating an autonomous mobile manipulator. Each case study will be thoroughly explained in its own subsection and will tackle one of the aforementioned contributions of this paper. Firstly, the Sensors to Odom Frame Test roBot 2 (SOFTBot2), a small simulated mobile robot with a robotic arm mounted on its platform, assesses the difficulty of calibrating the transformation between the platform and the base of the manipulator. This case study is more thoroughly explained, as it is also used to further explain the calibration pipeline in ATOM. Whilst this case study assumed perfect simulated odometry, the next case study features the Sensors to Odom Frame Test roBot (SOFTBot), a simpler version of the prior robot, without a robotic arm. In this case study, noise was artificially introduced in the geometric transformations provided by the simulated odometry system, thus making the estimation of these transformations also necessary. Finally, the last case study features *Zau*, a real mobile robotic manipulator, encompassing all the challenges of the other case studies along with the intricacies of a real system. It features sensors of three different modalities, more than one non-fixed sensor, and an odometry system with cumulative error.

In order to carry out these case studies, it was necessary to obtain sensorial data for each system. For each, a *ChAruco* chessboard was fixed on the environment and the robot moved between different poses. Table 1 represents a high-level view of the datasets collected along with a short description of each robotic system. A partial collection is a collection where at least one sensor did not fully detect the calibration target. Note that all collections recorded for *Zau* are partial because two of the RGB cameras have non-intersecting fields of view. In other words, either the pattern is partially detected on both cameras or is only fully detected on one.

### 4.3. Calibrating Mobile Manipulators

In order to assess the calibration of the transformation from the base platform to the manipulator of a mobile manipulator system, we will use a simulated robotic system called SOFTBot2, shown in Figure 1. This is a simulated system, equipped with five sensors: a *LiDAR*, a *body rgb camera*, and a *body depth camera,* mounted in the vehicle chassis, a *hand rgb camera* and a *hand depth camera*, assembled on the end effector of a robotic manipulator. The system is equipped with three sensor modalities, RGB, *LiDAR*, and depth, and contains two sensors in an eye-in-hand configuration. As such, it represents a complex calibration challenge, which includes elements of all the calibration problems discussed in Section 1: **sensor to sensor**, **sensor in motion**, and **sensor to frame**.

Figure 2 shows a schematic of the calibration procedure. The calibration pattern (*p*) is positioned in the scene and is not moved throughout the calibration session. Thus, in order to obtain different views of the pattern, the vehicle must move around. Moreover, the arm should also be positioned in different poses to achieve sufficient variability in the data used for calibration.

Figure 2 shows the relevant transformations in the context of the calibration of a mobile manipulator such as SOFTBot2. The root coordinate system of the vehicle is the vehicle base (*vb*). There are portions of the tree which represent the *LiDAR*, the *body camera*, and the *hand camera*. The scheme highlights the most important transformations for the calibration procedure, using two distinct notations. Tcba denotes a standard geometric transformation from coordinate frame *a* to frame *b*, at a particular collection *c*. This is a general concept, where *a* and *b* can represent any two coordinate frames in the system. The example shown in Figure 2 is the transformation between the arm base (ab) and the end effector (ee) of the robotic arm (Tc=0eeab), which is an aggregate transformation, as it comprises the combined motion of the shoulder, arm, forearm, wrist, and all other elements that compose the robotic arm. The other notation, Tcba, also represents a transformation from coordinate frame *a* to frame *b*, at a particular collection *c*. The difference here is that *a* and *b* must be topologically adjacent, thus encoding an atomic transformation.

As the vehicle moves during the calibration session, a localization system outputs estimates of its pose w.r.t. the world coordinate system, which are represented by the atomic transformations between the world (*w*) coordinate frame and the vehicle base (vb) frame, i.e., Tcvbw,∀c∈{0,1,…,n}. Any localization system can be used, but it is recommended that it provide accurate estimates, as these will influence the quality of the calibration. Each sensor in the system contains a coordinate frame to which the data it produces are referenced. For example, the *LiDAR* produces point cloud data referenced to the *lidar* (*l*) coordinate frame, the *body rgb camera* produces RGB images referenced to the *body rgb camera optical* (*brco*) coordinate frame, and the *body depth camera* produces depth maps referenced to the *body depth camera optical* (*bdco*) coordinate frame. For each sensor, an atomic transformation that encodes its pose w.r.t. the vehicle must be defined. These transformations are to be estimated by the calibration procedure, and we use the hat notation $\hat{T}$ to highlight it. The calibration of the *LiDAR* sensor is achieved by estimating the atomic transformation between the *lidar plate* (lp) and the *LiDAR* (*l*) coordinate frames, denoted as T^llp. The *hand rgb camera* undergoes calibration through the estimation of the transformation between the end effector (ee) and the *hand rgb camera* (hrc) frames, denoted as T^hrcee. Similarly, the *hand depth camera* is calibrated by estimating the transformation between the end effector (ee) and the *hand depth camera* (hdc) frames. Finally, the *body rgb camera* and the *body depth camera* utilize the mount plate (mp) to establish atomic transformations to the *body rgb camera* (brc) and *body depth camera* (bdc) frames. These transformations are denoted as T^brcmp and T^bdcmp, respectively.

To be selected as transformations to be estimated, these sensor transformations should, ideally, be static, meaning that the sensors are rigidly attached to the vehicle, or the end effector, in the case of the *hand rgb camera* or the *hand depth camera*. Ensuring this simplifies the optimization procedure by greatly reducing the number of parameters to be estimated. For this reason, sensor transformations do not contain the sub-index *c*. This particular issue hinders the calibration of mobile robots with imprecise localization systems, as these are not static and negatively affect the estimation of every other geometric transformation. The next contribution in the next case study assesses these difficulties.

Because this is a mobile manipulator, one essential transformation to estimate is one that positions the robotic arm w.r.t. the vehicle. This is denoted by the atomic transformation T^abvb, which is also static. Without knowing this transformation, while it is certain where the end effector is relative to the base of the manipulator, it is completely uncertain where the end effector is relative to the rest of the robot, which is typically necessary. Finally, it is also necessary to estimate the position of the calibration pattern in the scene, T^pw, as this pose is unknown.

Note that the description of the relevant atomic transformations in the paragraph above is only a typical setup for the calibration of a mobile robot. There are several alternatives, since it is possible to select different transformations to be estimated and still achieve valid calibrations. However, there are some good practices that are detailed next to achieve a better calibration. Considering sensors, for example, a sensor is independently positioned, as long as the selected transformation belongs to the transformation tree branch that is exclusive to the sensor. In the case of the *body rgb camera*, the selected transformation is T^brcmp, but it is possible to use T^brcobrc instead. However, it would not be ideal to use the T^mpvb transformation, as this would also affect the position of the *LiDAR*. Having one transformation impact the pose of two sensors creates redundancies that can push the optimizer towards local minima, and thus, a worse calibration. Furthermore, ATOM was designed to deal better with sparse optimization problems, so choosing to estimate a transformation that influences multiple sensors should negatively affect its performance.

Typically, we select the transformation which is closer to the sensor. The cameras, both RGB and depth, are an exception, because the final transformation in a camera is a standard transformation to swap from a classic x-forward, y-left, z-up, axis configuration to an optical standard z-forward, x-right, y-down configuration, which is used in cameras, as per the Robot Operating System (ROS) convention. Thus, these final transformations are known and standard, which is why it makes more sense to estimate instead the transformations which immediately precede those.

As mentioned before, in order to properly calibrate a mobile manipulator, it is needed to estimate Tabvb. In Equation (Equation 8), one can see that for any sensor modality, the error function always depends on the aggregated transformation from the sensor to the pattern Tps. For any sensor that is mounted on the robotic arm, this aggregated transformation always includes the atomic transformation from the vehicle base to the arm base Tabvb. This means that by estimating Tabvb, the optimization problem becomes less sparse, thus theoretically lowering the performance. Furthermore, as mentioned previously, having two transformations to calibrate one sensor leads to ambiguities, which in turn leads to local minima convergence and lower calibration performance.

In Table 2, the results for the calibration of SOFTBot2 are presented. The calibration performance was measured by the root-mean-square error, explained in Section 4.1. Both translation and rotation noises, with magnitudes of 0.1 m/0.1 rad, respectively, were added to all the atomic transformations being estimated. Comparisons with alternative calibration algorithms from *OpenCV* and Iterative Closest Point (ICP) calibration are also provided. All *RGB* cameras have a resolution of 1280 × 720p, while the depth cameras have a resolution of 640 × 480p. *OpenCV* has an implementation that is based on [48] and *Matlab’s Camera Calibration Toolbox.*

In Table 2, rows listing only ATOM represent sensor pairs with modalities beyond the capabilities of the other two approaches. Furthermore, some of the calibrations with the other methods were also not possible, as they violate some of the constraints those algorithms impose, such as partial detections or non-static sensors. This further emphasizes the versatile nature of ATOM, being able to deal with all those scenarios.

Regarding the results from ATOM, it is noteworthy that most calibrations achieve sub-pixel accuracy. The *RGB* to *RGB* calibration, in particular, reaches a notable 0.616 pix. To evaluate the quality of the transformation from the vehicle base, $v_{b}$, to the arm base, $a_{b}$, in Table 2, the metric presented in Section 4.1.2 was used, as no sensor data originate from these frames. Using this approach, ATOM achieved a translation error, $\epsilon_{t}$, of 0.005 m, and a rotation error, $\epsilon_{R}$, of 0.003 rad.

In this case study, it was shown that ATOM can now correctly estimate an arbitrary transformation, which can be useful in many scenarios, such as the one shown in this subsection. However, the simulated odometry system is assumed to be perfect. This is not representative of real-world systems, which inevitably exhibit imperfections. In the next case study, we demonstrate how ATOM can handle imprecise localization systems by introducing noise into the geometric transformations provided by such systems in a simplified robot model referred to as SOFTBot.

### 4.4. Calibrating Mobile Robots with Imprecise Localization Systems

The precision of the localization system directly impacts the quality of calibration achieved with the proposed approach. The cost function, defined in Equation (Equation 8), relies on the aggregated transformations from the sensors to the calibration target via the kinematic chain of the robot. The localization system provides the transformation between the origin frame and the base frame of the robot, which is part of every said aggregated transformation. Thus, inaccuracies in the transformation provided by the localization system adversely affect the cost function. SOFTBot, shown in Figure 3, was used to demonstrate how ATOM handles this issue. In the experiments presented in this subsection, translation and rotation noise with magnitudes of 0.1 m/0.1 rad, respectively, were added to all atomic transformations being estimated for each sensor, as well as to the atomic transformation provided by the localization system. Figure 4 shows the relevant transformations in the context of the calibration of SOFTBot, along with its sensor set.

In the previous case study, a single atomic transformation was estimated for each sensor undergoing calibration. This approach assumed that the remaining relevant transformations were known, allowing ATOM to compensate for minor uncertainties. However, depending on the accuracy of the localization system used, these uncertainties can become significant or even cumulative, making it necessary to address them explicitly during the calibration process. To evaluate this assertion, a calibration simulation was conducted with noise added to the geometric transformations provided by the simulated odometry system. Similarly to the transformations of the sensors, here the magnitude of the noise added to each component was also 0.1 m/0.1 rad, respectively. The results of this simulation are presented in Table 3. Note that the following results are **not** using the novel proposed approach.

The conclusion one can draw from Table 3 is that the uncertainties present in the transformations provided by the localization systems cannot be ignored, as they directly affect every other sensor being calibrated. The exact numerical values shown are not as important, as the inaccuracy of the localization system can vary.

In the previous case study, it was shown that ATOM can also estimate an arbitrary transformation, in addition to the transformations relating to each sensor. Previously, the transformation from the vehicle base, $v_{b}$, to the arm base, $a_{b}$, was this arbitrary transformation. Likewise, a robot with an inaccurate localization system has a similar calibration challenge: also calibrating the transformation provided by such a system. The difference here is that these transformations are dynamic. ATOM tackles this by estimating each transformation for each collection, as mentioned in Section 3.2. In contrast, other approaches opt to calibrate the odometry system itself, by estimating kinematic parameters such as wheelbase or wheel radius, like the works of [49] or [50]. The issue with these approaches is that they are specific for a certain type of locomotion mechanism, such as differential or Ackerman steering, and they are limited to odometry systems. On the other hand, our approach is completely agnostic to the localization system used and works for any type of locomotion mechanism.

To demonstrate these capabilities, a similar simulation to the one whose results are portrayed in Table 3 was conducted. However, now ATOM is also estimating the transformation provided by the odometry system for each collection. The results are shown in Table 4.

The results presented in Table 4 demonstrate the efficacy of ATOM in handling the calibration of robotic systems with imprecise localization systems. By effectively estimating the transformations provided by the odometry system, ATOM can successfully calibrate the system. Given a certain dataset of size *n*, this approach requires the calibration of *n* additional transformations. There are no other methods presented both in Table 3 and Table 4, as to the best of our knowledge, there are no other works that can simultaneously deal with inaccurate odometry and calibrate a mobile robot with multiple sensor modalities.

Similarly to the last case study, it is not possible to evaluate the quality of the calibration of the transformation provided by the odometry system in the same manner, as there are no sensorial data coming from those frames, and thus, the metric presented in Section 4.1.2 was used again. Using this metric, our approach achieved an average translation error, $\epsilon_{t}$, and an average rotation error, $\epsilon_{R}$, of 0.0127 m and 0.0119 rad, respectively. The standard deviations for these averages are 0.0192 m and 0.0179 rad. These deviation values are high, as the sample only has 4 elements, with one transformation per sensor being calibrated, and the transformation provided by the odometry, and LiDAR calibration poses many additional challenges [51], thus making it less precise than camera to camera calibration and skewing the standard deviation towards a higher value. The individual errors per transformation being estimated are shown in Table 5, in which one can see the aforementioned higher magnitude of the LiDAR errors. Additionally, the error on the transformation from the vehicle base to the left camera is null, as this transformation is anchored, thus remaining fixed throughout optimization. This technique is otherwise known as *Gauge Freedom* [52] and helps with systems that are not fully constrained.

### 4.5. Calibrating a Real Mobile Manipulator with Imprecise Odometry

Whilst the last two case studies show how ATOM can calibrate a mobile manipulator, many factors are not accounted for in simulation. If ATOM could not also calibrate such a robotic system in a real life scenario, the viability of the framework would be fundamentally limited. With this in mind, this last case study features a real life mobile manipulator named *Zau*. It features two RGB cameras with non-overlapping fields of view, a 3D LiDAR rigidly attached to the body and an RGBD camera attached to the end effector of the manipulator. *Zau* features a built-in proprietary odometry system. A figure of the experimental setup containing *Zau* can be seen in Figure 5, and the relevant transformations in the context of the calibration of *Zau* can be seen in Figure 6.

The calibration of this robotic system entails all the challenges of the previous case studies, alongside those derived from being a real system. Firstly, like in the calibration of SOFTBot2, it is necessary to estimate the transformation from the vehicle base, $v_{b}$, to the arm base, $a_{b}$. Additionally, it is also necessary to estimate the transformations provided by the odometry system for each collection, like in the calibration of SOFTBot. These challenges make calibrating a robotic system such as *Zau* a complex problem.

In order to collect a dataset required for the calibration procedure, a *ChArUco* chessboard was attached on a tripod and fixed in place in the environment. During the calibration session, both the base of robot and the manipulator attached to the top were moved throughout a variety of poses, and collections are recorded at each desired pose. Having a wide variety of poses heavily reduces the risk of falling into local minima during optimization, as there are fewer ambiguities. Afterward, the calibration procedure was carried out, ensuring the estimation of one transformation per sensor being calibrated; the transformation from the world to the frame of the calibration pattern; each transformation provided by the odometry system; and finally, the transformation from the base of the platform to the base of the robotic manipulator. The results from this calibration are shown in Table 6 and Table 7.

From Table 6 and Table 7, one can see that most evaluations achieve a sub 10 pixel root-mean-square error. While ATOM has achieved greater results in the calibration of other robots [14], it was always in much simpler robotic systems. To our knowledge, none of the aforementioned alternative methodologies can calibrate a system such as *Zau*, either because the collections have partial detections, or mainly because the sensors are not static. Adding to this, no other known methodology can simultaneously deal with an inaccurate localization system; calibrate arbitrary transformations such as the one from the base of the platform to the base of the manipulator; and extrinsically calibrate all onboard sensors with different modalities. These results also highlight the robustness of the framework, as ATOM can provide accurate calibrations despite the obstacles imposed by transposing the calibration problem from simulation to a real system.

## 5. Conclusions

In this paper, we have advanced the capabilities of the previously established ATOM framework to address the multifaceted calibration challenges presented by mobile manipulators. We showed how our approach can cope with the calibration of arbitrary transformations, be they static, such as in the case of transformation from the vehicle base to the arm base in SOFTBot2 or *Zau*, or even dynamic, such as in the case of the transformation provided by the odometry system in SOFTBot or in *Zau*. ATOM shows resilience to common calibration limitations, such as handling partial detections, dynamic transformations, and non-overlapping sensor fields of view. By not showing a significant drop in performance in real-world settings, ATOM is also a great and viable option to implement in production systems as of right now.

Future work should delve into exploring the calibration of more sensor modalities, such as Inertial measurement units (IMUs), and multi-robot calibration. Together with the contributions of this paper, multi-robot calibration would allow ATOM to endeavor into the real of humanoid robot calibration. After all, a humanoid robot can be simplified to a mobile robot with two robotic manipulators and an exquisite locomotion mechanism.

## Figures and Tables

**Figure 1 sensors-25-02501-f001:**
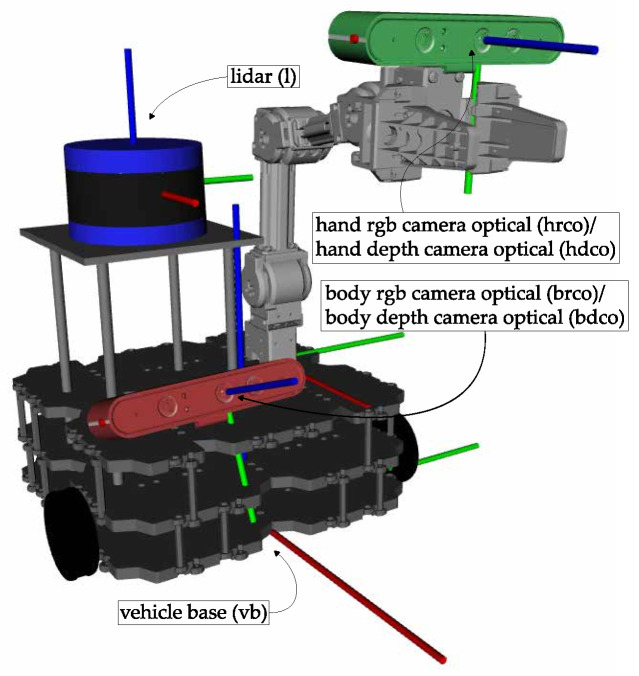
The SOFTBot2 robotic system, a simulated mobile manipulator containing three sensors: a *body camera* (red), a *hand camera* (green), and a *lidar* (blue). The coordinate frames of these sensors are displayed, in addition to the vehicle base and the manipulator base frames. The RGB and Depth optical frames of both cameras are separated by a fixed offset. They are represented in a simplified manner to improve figure clarity.

**Figure 2 sensors-25-02501-f002:**
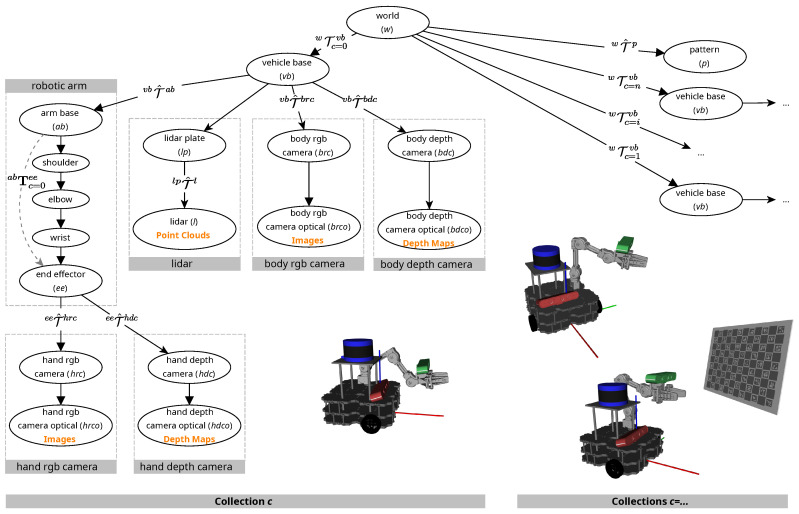
Relevant transformations in SoftBOT2. On the left, a full transformation tree is shown with boxes surrounding elements of interest for a given collection *c*. On the right, a hint of the trees for subsequent collections is visible, highlighting that the world frame is fixed and transformations between collections can be obtained. The various renders of SoftBOT2 draw attention to the fact that collections can often have completely different system states.

**Figure 3 sensors-25-02501-f003:**
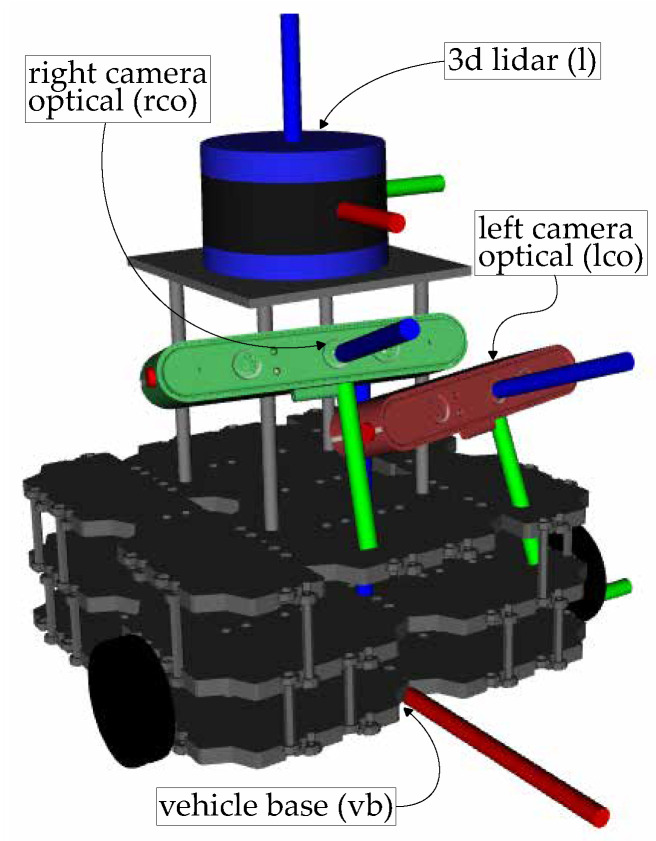
The *SOFTbot* robotic system, a simulated mobile robot containing three sensors: a *left camera* (red), a *right camera* (green), and a *3D LiDAR* (blue). The coordinate frames of these sensors are displayed, in addition to the vehicle base.

**Figure 4 sensors-25-02501-f004:**
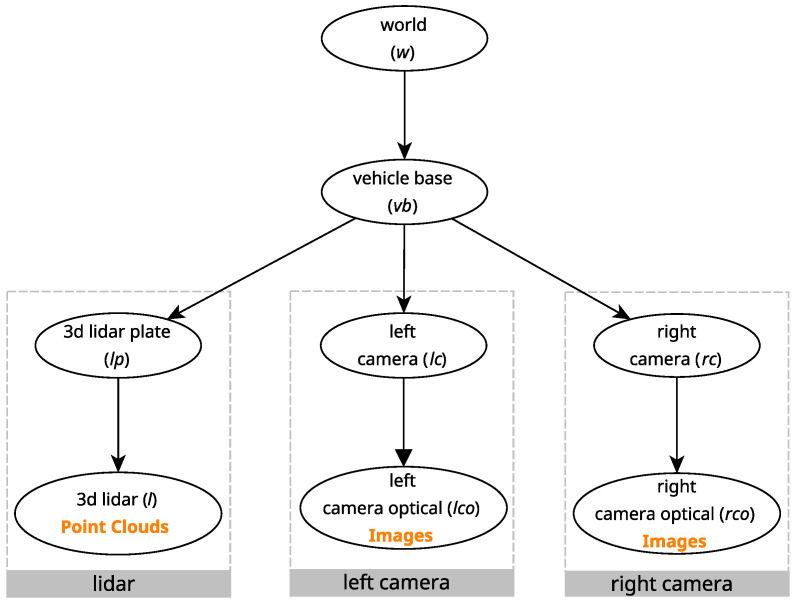
Relevant transformations in *SoftBOT*. The dotted gray boxes contain internal transformations within a certain sensor. The coordinate frames that generate data have the type of data they generate in orange below.

**Figure 5 sensors-25-02501-f005:**
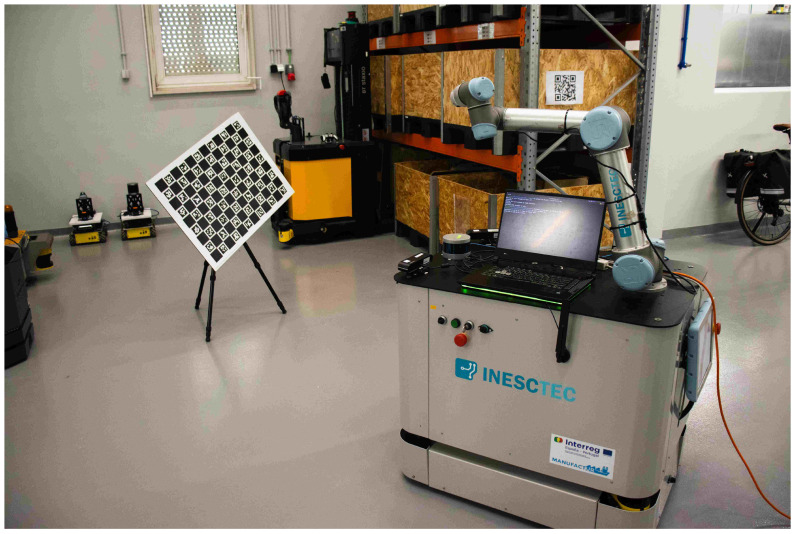
The *Zau* robotic system, a real mobile manipulator containing four sensors: two *RGB body cameras*, an *RGBD hand camera*, and a *LiDAR body*.

**Figure 6 sensors-25-02501-f006:**
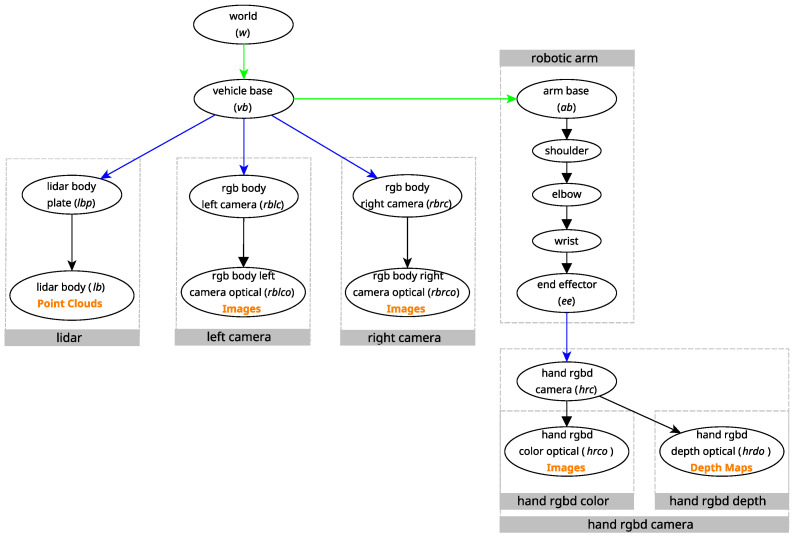
Relevant transformations in *Zau*. The dotted gray boxes contain internal transformations within a certain sensor. The coordinate frames that generate data have the type of data they generate in orange below. Transformations that could be calibrated using our previous method are marked in blue, whereas the transformations newly supported in this work are shown in green.

**Table 1 sensors-25-02501-t001:** Robotic systems used to evaluate ATOM as well as the datasets used for train and test.

System	Description	Sensors	Dataset	Details
SOFTBot ^(a)^	Mobile robot Differential steering	RGB cameras (2×) 3D LiDAR	Simulated	44 collections, 3 partial
SOFTBot2 ^(b)^	Mobile manipulator Differential steering	RGBD cameras (2×) 3D LiDAR	Simulated	54 collections, 36 partial
Zau ^(c)^	Mobile manipulator Differential steering	RGB cameras (3×) 3D LiDAR Depth camera	Real	50 collections, 50 partial

^(a)^ https://github.com/miguelriemoliveira/softbot (accessed on 1 February 2025); ^(b)^ https://github.com/miguelriemoliveira/softbot2 (accessed on 1 February 2025); ^(c)^ https://github.com/lardemua/zau (accessed on 1 February 2025).

**Table 2 sensors-25-02501-t002:** Simulation performance comparison of camera evaluation methods calibrating SOFTBot2. Best values in bold.

Method	Sensor Pair	$\epsilon_{\mathrm{rms}}$
		(pix)
OpenCV	*body rgb camera* to *hand rgb camera*	^(a)^
ATOM		**0.616**
ATOM	*3d lidar* to *body rgb camera*	2.367
ICP		2.317
ATOM	*body depth camera* to *body rgb camera*	1.466
	*body depth camera* to *hand rgb camera*	1.495
	*hand depth camera* to *body rgb camera*	2.402
	*hand depth camera* to *hand rgb camera*	2.472
ATOM	*body depth camera* to *3d lidar*	**1.534**
ICP average		15.136
ICP best		53.644
ATOM	*hand depth camera* to *3d lidar*	**1.580**
ICP average		^(b)^
ICP best		^(b)^
ATOM	*body depth camera* to *hand depth camera*	**1.457**
ICP average		^(b)^
ICP best		^(b)^

^(a)^ Method cannot be used because the dataset contains partial detections. ^(b)^ Method cannot be used because the one of the sensors is not static.

**Table 3 sensors-25-02501-t003:** Simulation performance of *SoftBOT* without correcting the transformation provided by the odometry system.

Method	Sensor Pair	$\epsilon_{\mathrm{rms}}$
		(pix)
ATOM	*left camera* to *right camera*	50.854
	*3d lidar* to *left camera*	55.858
	*3d lidar* to *right camera*	55.148

**Table 4 sensors-25-02501-t004:** Simulation performance of *SOFTBot*.

Method	Sensor Pair	$\epsilon_{\mathrm{rms}}$
		(pix)
ATOM	*left camera* to *right camera*	0.243
	*3d lidar* to *left camera*	6.499
	*3d lidar* to *right camera*	7.131

**Table 5 sensors-25-02501-t005:** Simulation performance of *SOFTBot*: absolute metric error.

Transformation	$\epsilon_{t}$	$\epsilon_{R}$
	(m)	(rad)
*vehicle base* to *left camera*	0	0
*vehicle base* to *right camera*	$8.26\times 10^{-3}$	$6.10\times 10^{-4}$
*vehicle base* to *3d lidar plate*	$1.97\times 10^{-2}$	$2.81\times 10^{-3}$
*world* to *vehicle base*	$5.08\times 10^{-2}$	$4.24\times 10^{-2}$

**Table 6 sensors-25-02501-t006:** Real performance of *Zau*: results containing *RGB* and *Depth* sensors.

Sensor Pair	*rgb Body Left* to *rgb Body Right*	*rgb Body Left* to *rgbd Hand Color*	*rgb Body Right* to *rgbd Hand Color*	*rgbd Hand Depth* to *rgb Body Left*	*rgbd Hand Depth* to *rgb Body Right*	*rgbd Hand Depth* to *rgbd Hand Color*
$\epsilon_{rms}$ (pix)		10.127	5.841	6.580	8.151	5.799

Evaluation cannot be computed because fields of view of the sensors in the pair do not overlap.

**Table 7 sensors-25-02501-t007:** Real performance of *Zau*: results containing *LiDAR* sensors.

Sensor Pair	*Lidar Body* to *rgb Body Right*	*Lidar Body* to *rgb Body Left*	*Lidar Body* to *rgbd Hand Color*	*Lidar Body* to *rgbd Hand Depth*
$\epsilon_{rms}$ (pix)	8.247	4.591	10.268	7.313

## Data Availability

All data and source code are available in the GitHub repositories listed throughout the article: https://github.com/lardemua/atom (accessed on 1 February 2025); https://github.com/miguelriemoliveira/softbot (accessed on 1 February 2025); https://github.com/miguelriemoliveira/softbot2 (accessed on 1 February 2025); https://github.com/lardemua/zau (accessed on 1 February 2025).

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
