# Peer review of "Calibration of Mobile Robots Using ATOM"

_sensors, 2025, doi:10.3390/s25082501_

Round 1
Reviewer 1 Report
Comments and Suggestions for Authors
sensors-3540851
Calibration of Mobile Robots using ATOM
This paper proposes a calibration approach that simultaneously estimates both the transformations provided by the localization system and the transformations between sensors and the motion coordinate system, enabling precise calibration even when the localization system is imprecise. The proposed methodology is validated through simulations and real-world case studies, demonstrating its effectiveness in improving calibration accuracy for mobile manipulators equipped with diverse sensor modalities. In general, the quality of this article is very good, and it can be published directly on sensors after modification. I mainly focus on the following minor issues.
- Some transformation matrices in Figure 2 are very fuzzy. It is recommended that the author optimize the quality of Figures.
- References in the past three years account for a small proportion. It is recommended that the author replace them appropriately. In addition, it is recommended to add excellent papers on sensors as references.
- I think the names of the coordinate systems and their respective three-dimensional coordinate systems should be given in Figures 1 and 3.
- In any case, the layout of this paper needs to be further optimized, and try not to have the situation of principle text description like Figure 6.
- It is recommended that the author compare them with the current advanced algorithms. The lack of comparison with advanced algorithms makes it difficult for reviewers and other readers to clearly understand the effectiveness and superiority of the proposed algorithm.
Author Response
Dear Reviewer 1,
Please see the attachment.
Changes in the manuscript are marked in blue.

Reviewer 2 Report
Comments and Suggestions for Authors
In order to solve the multifaceted calibration challenges presented by mobile manipulators, the authors improved the capabilities of the previously established ATOM framework. The simulation performance comparison verifies the superiority of the improved calibration method, and the calibration results of the real mobile manipulator also verify the effectiveness of the improved method.
In order to better verify the effectiveness and superiority of the proposed improved calibration method, it is recommended to compare the performance of different calibration methods for real mobile robotic manipulator.
In line 193, the variable dcs is not found in equation (4).
The English could be improved to more clearly express the research.
Author Response
Dear Reviewer 2,
Please see the attachment. The answer to one of your questions is on the reviewer's 1 report as you shared the same concern, it is properly detailed on your report where to check.
Changes in the manuscript are marked in blue.

Reviewer 3 Report
Comments and Suggestions for Authors
The article is devoted to solving the problem of calibration of a mobile manipulative robot.
А method for estimation the transformation between the sensors and the motion coordinate system, for estimation the transformations given by inaccurate localization systems is proposed for a complex real mobile manipulator with many sensors.
The article contains links to github repositories of software for solving calibration tasks. Its results can be used to control the movement of manipulative robots, mobile robots, anthropomorphic robots, etc.
There are a number of comments on the article's material.
It is necessary to indicate in the text or add references to the literature to explain what intrinsic parameters λ (line 202), matrix K (line 390) are.
Formulas (14)-(17) use the parameters R, t. What they are is not specified.
Lines 379, 409. It is not specified how the angle( ) calculation is performed in axes-angle representation.
In section 4, error values ϵt, ϵr, and ϵrms are given for comparing methods. Has algorithmic efficiency been investigated for the calibration methods considered? Does the calibration time limit the robot's movement speed?
Line 611. The average values of errors ϵt, ϵr are given. To understand the accuracy of the method, it is necessary to specify the standard deviation and the magnitude of the errors.
There are a few editorial comments. The text uses different designations for the same quantities and concepts: ZAU, Zau; ϵt, ϵT; ϵr, ϵR.
Line 574. ϵt = 0.003 rad instead of ϵr = 0.003 rad
I recommend that the authors add the 2024 article to the review:
Sumenkov O. Y., Kulminskiy D. D., Gusev S. V., Kinematic Calibration of an Industrial Manipulator without External Measurement Devices, Rus. J. Nonlin. Dyn., 2024, Vol. 20, no. 5, pp. 979-1001
DOI:10.20537/nd241003
The article can be published after the removal of these comments.
Comments on the Quality of English LanguageIn my opinion, the quality of the English in the article is quite high.
Author Response
Dear Reviewer 3,
Please see the attachment.
Changes in the manuscript are marked in blue.

Reviewer 4 Report
Comments and Suggestions for Authors
The proposed paper proposes an optimization method for mobile manipulators by extending the ATOM method. However, since most manipulators provide their own accurate odometry, it is believed that the existing ATOM method can be sufficiently calibrated. In addition, the results of the proposed method are different from the results published by ATOM, in that only the analysis of pixels exists. The numerical value of pixels is difficult to view as a calibration result. This is because the accuracy is ambiguous in the world coordinate system.
Author Response
Dear Reviewer 4,
Please see the attachment.
Changes in the manuscript are marked in blue.

Round 2
Reviewer 1 Report
Comments and Suggestions for Authors
Accept
Author Response
Dear Reviewer 1,
Thank you for accepting our paper.
Reviewer 4 Report
Comments and Suggestions for Authors
The research has discussed a case in which the previously proposed correction method (14) was applied to a mobile manipulator and verified.
I think the originality of the paper is very low. The ATOM concept mentioned in the 14th paper was directly borrowed and applied to the mobile manipulator, but compared to the methods mentioned in the existing papers, I think the novelty is very lacking. If you compare the formulas, they are very similar in terms of method. For example, while the correction for two modalities in the 14th paper was fixed and the relative error was formulated, the proposed paper includes the distance between the two in the error.
Furthermore, the ATOM(14) correction shows results for mobile or manipulators, while the proposed paper adds that it is applied to mobile manipulators. This means that it is not specialized for a specific type of movement mechanism but is applicable to all types of movement mechanisms.
The authors added a method to correct the added modality based on odometry. However, it is difficult to judge the usefulness of the paper only with the results presented in Tables 3 and 4, as it is a very simple method to compare the results of adding noise to the existing method and correcting it with the odometry system to show the usefulness of the proposed method.
The presented claim shows the result of ϵr of 0.0127 m and 0.0119 rad through simulation comparison. It is difficult to judge the effectiveness only with the results mentioned in the text. Detailed comparison results, such as those in paper 14, are required.
Thus, as in paper 14, a large amount of data needs to be quantified. Since it is difficult to estimate the error in the world coordinate system with pixels alone, the results need to be expanded.
Author Response
Dear Reviewer 4,
Please find attached the notes for this second round of revisions.
Best Regards,
Bruno Silva
